# A double-sided microscope to realize whole-ganglion imaging of membrane potential in the medicinal leech

**Yusuke Tomina\*, Daniel A Wagenaar**

Division of Biology and Biological Engineering, California Institute of Technology, Pasadena, United States

**Abstract** Studies of neuronal network emergence during sensory processing and motor control are greatly facilitated by technologies that allow us to simultaneously record the membrane potential dynamics of a large population of neurons in single cell resolution. To achieve whole-brain recording with the ability to detect both small synaptic potentials and action potentials, we developed a voltage-sensitive dye (VSD) imaging technique based on a double-sided microscope that can image two sides of a nervous system simultaneously. We applied this system to the segmental ganglia of the medicinal leech. Double-sided VSD imaging enabled simultaneous recording of membrane potential events from almost all of the identifiable neurons. Using data obtained from double-sided VSD imaging, we analyzed neuronal dynamics in both sensory processing and generation of behavior and constructed functional maps for identification of neurons contributing to these processes.
DOI: https://doi.org/10.7554/eLife.29839.001

**\*For correspondence:**
tominaye@caltech.edu

[†]Please contact daw@caltech. edu (to DW) or homarus. tomina@gmail.com (to YT) when tominaye@caltech.edu is unavailable.

**Competing interests:** The authors declare that no competing interests exist.

## Introduction

One of the principal goals in neuroscience is to clarify how neuronal circuits process sensory information and control behavior. Sensory information and behavioral states are represented as dynamic activity patterns of neuronal populations in large neuronal networks. To clarify the neuronal mechanisms underlying sensory processing and behavioral generation, it is necessary to determine which neurons are involved in functionally relevant neuronal dynamics and how those neuronal components interact with each other within the larger network. Technological advances in neuroimaging have enabled brain-wide recording of neuronal activity with sufficiently fine spatial resolution to identify individual neurons within a population (*Ahrens et al., 2013*). Researchers can perform pan-neuronal $Ca^{2+}$ imaging in selected animals with nervous systems comprising small neuronal populations, including larval zebrafish (*Ahrens et al., 2013*) and *C. elegans* (*Schrödel et al., 2013*; *Kato et al., 2015*; *Nguyen et al., 2016*).

Although $Ca^{2+}$ imaging is a convenient tool for detecting neuronal activity, it is limited to intracellular events that are associated with a change in $Ca^{2+}$ concentration. Thus, $Ca^{2+}$ imaging measures neither subthreshold depolarizing nor hyperpolarizing synaptic events. Accordingly, it is difficult to observe synaptic integration processes using $Ca^{2+}$ indicators. In contrast, voltage sensitive dyes (VSDs) can detect both action potentials and sub-threshold excitatory and inhibitory synaptic potentials. Voltage sensors have enabled neuroscientists to examine ethologically relevant neuronal dynamics and to functionally map parts of the nervous systems of sea slugs (*Bruno et al., 2015*; *Hill et al., 2015*; *Hill et al., 2014*) and the medicinal leech *Hirudo verbana* (*Briggman et al., 2005*; *Briggman and Kristan, 2006*; *Frady et al., 2016*). The central nervous system of the leech consists of a ventral nerve cord connecting a head brain, 21 nearly identical segmental ganglia and a tail brain. The segmental ganglion of the leech is particularly well suited for comprehensive recording

**eLife digest** In every animal, networks of nerve cells work together to interpret signals from the environment and to coordinate responses. Being able to record the activity of all the neurons in a brain at once would greatly advance our understanding of how the brain works. Yet it is not possible to do this for a human brain, which contains several billion neurons. The medicinal leech, on the other hand, has a much simpler nervous system. It has 21 brain-like units called segmental ganglia, which control how the parts of its body move, and each one contains about 400 neurons arranged on a single layer.

The activity of large populations of neurons can be monitored using a technique called fluorescent imaging. Most fluorescent dyes, however, are not sensitive enough to report low levels of activity or fast enough to track individual nerve impulses. Also, current microscopy techniques only allow one surface to be imaged at any one time. These limitations constrain the kinds of questions that neuroscientists can ask about how networks of nerve cells function.

Tomina and Wagenaar have now developed a double-sided fluorescent microscope system that allows a ganglion in a medicinal leech to be viewed from both sides at once. Using a new generation of dyes, which rapidly change their brightness as individual neurons become active or are inhibited, subtle changes in the activity of hundreds of individual neurons were monitored at the same time. In a test of the system, Tomina and Wagenaar recorded activity for different leech behaviors, like bending, swimming and crawling. For the first time, the relationships between neurons on both sides of the ganglion could be seen.

This new technique for examining the activity in neuronal circuitry will allow complex networks of neurons to be studied in more detail. The data that these images generate could then be analyzed mathematically to better understand how the brain processes information from its senses and generates behavior.

DOI: https://doi.org/10.7554/eLife.29839.002

using VSD imaging for two reasons: It consists of only about 400 identifiable neurons (*Pipkin et al., 2016*), mostly as bilateral pairs, arranged in a well-preserved geometry in a single spherical shell surrounding a central neuropil, and it functions as a basic unit of sensory processing and control of several behaviors (*Kristan et al., 2005*). In the leech segmental ganglion, multiple neuronal circuits responsible for reflexive and voluntary locomotor behaviors have already been characterized by electrophysiology and VSD imaging (*Briggman et al., 2005*; *Briggman and Kristan, 2006*; *Frady et al., 2016*; *Kristan et al., 2005*). However, existing technology only allowed imaging one side of a ganglion at a time, and hence captured the activity of at most half of the full ensemble of neurons: researchers could record from, at most, approximately 15% of all 6000 neurons in a pedal ganglion of *Aplysia*, and fewer than 50% of all 400 neurons in a leech segmental ganglion. In addition, VSDs applied in these previous studies were limited by their sensitivity or response speed: electrochromic dyes used in sea slugs did not possess enough sensitivity to detect subthreshold potentials (*Bruno et al., 2015*; *Hill et al., 2015*) and FRET-based dye previously used in the leech had a non-negligible delay between optical signal and actual voltage change (*Briggman et al., 2005*; *Briggman and Kristan, 2006*).

To overcome these limitations, we developed a double-sided microscope for VSD imaging, consisting of precisely aligned upright and inverted fluorescent microscopes, and imaged voltage changes from the neuronal membrane stained by a highly-sensitive, fast VSD (*Woodford et al., 2015*). This microscope enabled us to record from all cell bodies of a leech ganglion regardless of their location, and allowed us, for the first time, to directly analyze functional relationships between neurons located on opposite surfaces. We combined this double-sided neuronal imaging system with simultaneous electrophysiological recording and stimulation, which allowed us to monitor motor outputs, to verify agreement of VSD signals with actual membrane potentials, and to activate or inhibit selected target cells by injecting current.

To demonstrate the utility of the newly developed VSD imaging method, we addressed the following two questions. (1) How are individual identifiable neurons that exhibit higher discriminability

for the different sensory stimuli distributed across different surfaces of the ganglion? (2) To what extent are neural circuit components unique or shared between different behaviors?

## Results

### VSD imaging using double-sided microscopy system

Double-sided VSD imaging requires simultaneously focusing two fluorescent microscopes. We achieved this by mounting the fluorescence train of an Olympus BX upright microscope with a custom focus rack on top of the body of an Olympus IX inverted microscope. Both microscopes were equipped with 20x objectives. An optically stabilized high-power LED (*Wagenaar, 2012*) provided excitation light through the top objective, which operated in epifluorescence mode. The top objective also functioned as a condenser lens for imaging with the bottom objective, which thus operated in transfluorescence mode (*Figure 1a*). Because of the high NA (1.0) of the top objective, inhomogeneities in the imaged tissue did not cause substantive deviations from uniform illumination of the bottom focal plane.

The two microscopes were first coarsely aligned (to within about 200 μm) by moving the upright microscope's body and its objective turret, after which micro-alignment was achieved by fine-tuning the position of the upright microscope's objectives in their turret. We used highly sensitive CCD cameras (Photometrics QuantEM 512SC) to image neuronal activity with single cell resolution throughout the ganglion (*Figure 1b*). We suppressed mechanical vibration noise by replacing the internal fans of the CCD cameras with external blowers. Photon noise was not substantially different between the top and the bottom image (Top: 72 ± 3 ppm; Bottom: 65 ± 3 ppm (mean ± SEM over 10 areas size-matched to typical cells).

We imaged neural activity with a new-generation voltage sensitive dye, VF2.1(OMe).H (*Woodford et al., 2015*), which is sensitive enough to record subthreshold events and fast enough to detect action potentials with accurate timing. The dye was loaded into somatic membranes on both aspects of a ganglion by bath application and a perfusion pump for targeted delivery (*Briggman et al., 2005*). In leech ganglia, the sensitivity reached 2.7 ± 0.3 % per 100 mV (mean ± SD across five ganglia in two leeches) at resting potential (−50 mV) (*Figure 1—figure supplement 1*). Microscopic motion artifacts can have outsized effects on VSD signals compared to $Ca^{2+}$ signals because of the limited relative change in fluorescence of VSDs and their location in the cell membrane. Accordingly, we applied a custom motion correction algorithm to all imaging data (*Figure 1—figure supplement 2* and Materials and methods). Bleaching artifacts in the optical signals were corrected using locally fitted cubic polynomials (*Wagenaar and Potter, 2002*) (*Figure 1—figure supplement 3*) and global fluctuations were subtracted away (*Lippert et al., 2007*) (Materials and methods). The voltage sensor faithfully detected various types of membrane potential change, including action potentials, excitatory and inhibitory postsynaptic potentials, and rhythmic oscillation during fictive behaviors (*Figure 1d*). All motor patterns recorded in this study are fictive patterns.

### Panneuronal VSD imaging and functional mapping based on coherence analysis

We established a mapping between cells seen in the fluorescent images (*Figure 1b*) and identified neurons on a canonical map (*Figure 1c*) using a semi-automated procedure in a custom user interface (Materials and methods). One of the major advantages of VSDs is that recorded traces can be directly compared to intracellular voltage recordings. This allowed us to identify selected cells in our recordings by comparing our data to previously published intracellular activity of those neurons in the same behaviors. In our setup, the preparation is accessible to intracellular and extracellular electrodes advanced from the upper surface.

Optically recorded signals simultaneously recorded from both sides of the ganglion closely matched typical patterns of fictive behaviors that have been previously well characterized by electrophysiology and single-sided VSD imaging (*Briggman and Kristan, 2006*; *Kristan et al., 2005*). We first focused on fictive swimming, which we induced by electrically stimulating a DP nerve root of a posterior ganglion (*Briggman and Kristan, 2006*) (typically, the 13th segmental ganglion (M13)). We then imaged ganglion M10 with our double-sided microscope and simultaneously recorded intracellularly from selected cells (*Figure 2a*). Rhythmic activity associated with swimming was readily

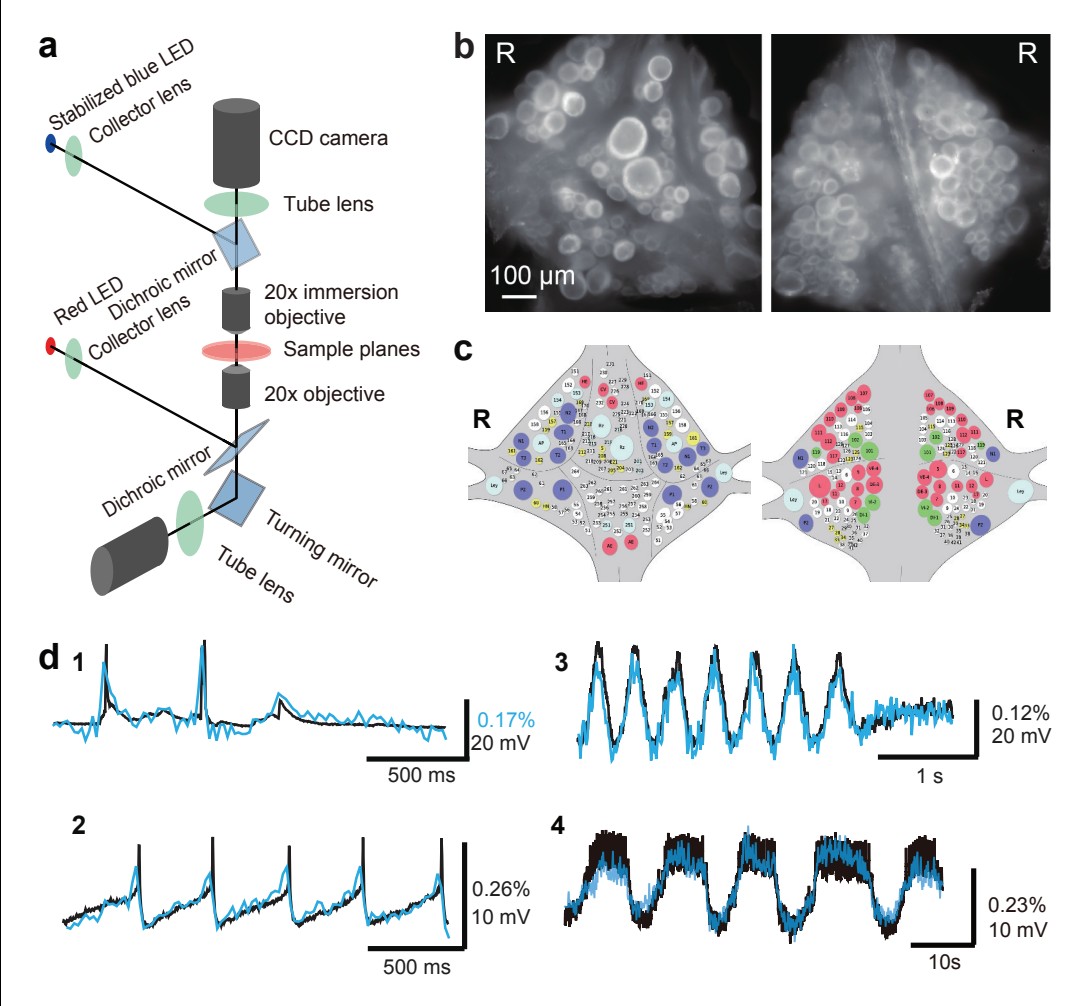

**Figure 1.** Double-sided voltage sensitive dye imaging. (**a**) Schematic of the double-sided microscope. (**b**) Images of the ventral (*left*) and dorsal (*right*) aspects of a leech ganglion simultaneously acquired using this microscope. 'R' indicates the right side of the ganglion. ('Right' is the animal's right side when oriented dorsal side up) (**c**) Canonical maps of the ventral (*left*) and dorsal (*right*) aspects of the ganglion. (**d**) Single-sweep recordings of neuronal activity. Optical signals from VSD imaging (*blue*) are overlaid with simultaneous intracellular recordings (*black*). 1. Action potentials and subthreshold potentials in a Retzius cell; 2. Spontaneous regular firing in an AP cell; 3. Swimming pattern in a DE-3 motor neuron; 4. Crawling pattern in an AE cell.

DOI: https://doi.org/10.7554/eLife.29839.003

The following figure supplements are available for figure 1:

**Figure supplement 1.** Measurements of the sensitivity of the VoltageFluor voltage-sensitive dye VF2.1(OMe).H in a leech Retzius cell.

DOI: https://doi.org/10.7554/eLife.29839.004

**Figure supplement 2.** Correction of micro-motion in images.

DOI: https://doi.org/10.7554/eLife.29839.005

**Figure supplement 3.** Debleaching VSD signals by local curve fitting.

DOI: https://doi.org/10.7554/eLife.29839.006

observed, and we determined which cells were involved in this rhythm by calculating the phase and magnitude of coherence (*Briggman and Kristan, 2006*) for each cell at the frequency with the greatest spectral power in the rhythm (*Figure 2b,c*). The optical signal of dorsal inhibitor motor neuron DI-1 exhibits a well-understood swimming oscillation and was used as the phase reference for other cells. Using the VF2.1(OMe).H dye, we were able to confirm the oscillatory behavior of neurons

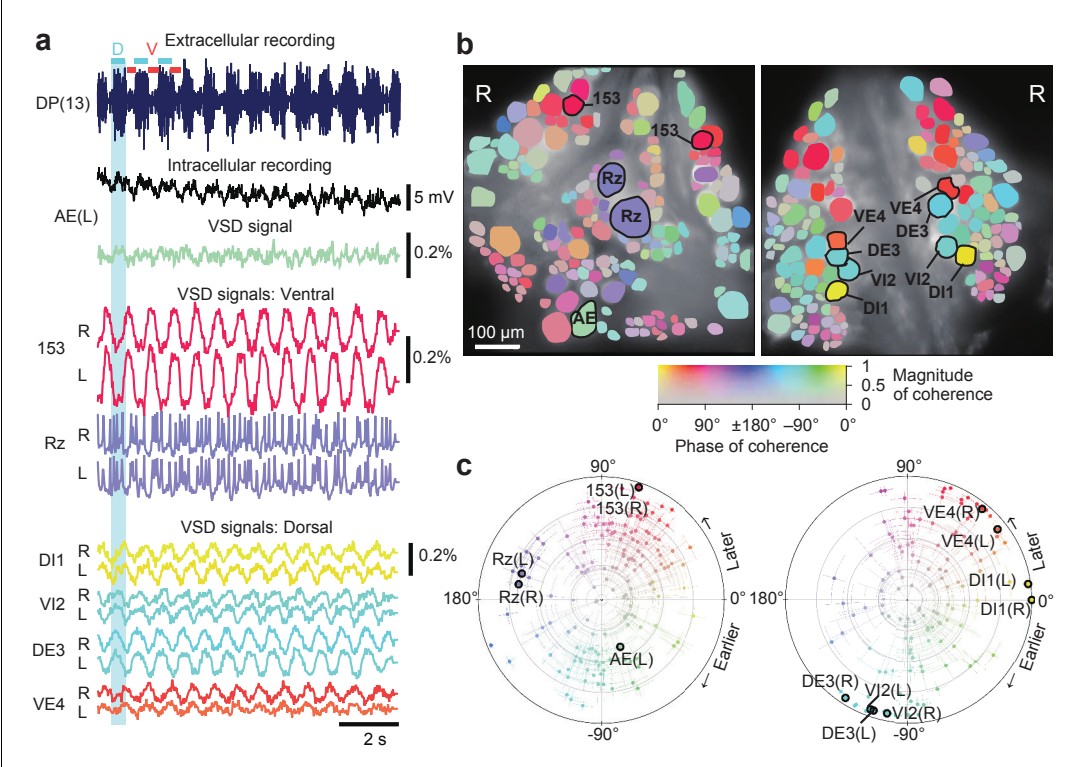

**Figure 2.** Neuronal activity during fictive swimming. (a) Selected electrophysiological and VSD traces during fictive swimming. Extracellular recording from a nerve root in a posterior segment (DP(13)) showed rhythmic dorsal motor neuron bursts characteristic of swimming (*top*). Intracellular recording and simultaneous optical signal from an AE neuron show matching membrane potential oscillations. VSD signals from the ventral surface: bilateral cells 153 (a sensory neuron) and the Retzius cell (a neuromodulatory neuron). VSD signals from the dorsal surface: dorsal and ventral inhibitory and excitatory motor neurons DI-1, VI-2, DE-3, and VE-4. (b) Coherence of the optically recorded signals of all cells on the ventral (*left*) and dorsal (*right*) surfaces of the ganglion with the swim rhythm. Cells used in (a) are marked. (c) Magnitude (radial axis from 0 to 1) and phase (angular coordinate) of the coherence of each neuron's activity with the swim rhythm; same data as in (b). Error bars indicate confidence intervals based on a multi-taper estimate. Data available from the Dryad Digital Repository: https://doi.org/10.5061/dryad.m20kh/7 (title: Figure 2, *Tomina and Wagenaar, 2017*).
DOI: https://doi.org/10.7554/eLife.29839.007

The following figure supplement is available for figure 2:

**Figure supplement 1.** Comparison of fictive swimming between single- and double-sided imaging.
DOI: https://doi.org/10.7554/eLife.29839.008

previously studied using an earlier-generation dye (*Briggman and Kristan, 2006*). In addition, we were able to detect weaker oscillations in many other neurons on both sides of the ganglion.

Results from coherence analysis obtained from doubly desheathed ganglia imaged using either camera in our double-sided microscope closely matched results from conventional single-sided imaging, as evidenced by the consistency of the coherence maps computed from either method (*Figure 2b* and *Figure 2—figure supplement 1*). The measured amplitudes of swim oscillations in motor neuron DI-1, the noise levels in those recordings, and the coherence between bilateral homologues of DI-1 were also indistinguishable between single-sided and double-sided imaging experiments (*Figure 2—figure supplement 1*), indicating that double-sided imaging does not entail any compromises from an imaging quality perspective.

## Encoding of stimulus identity by individual neurons

We used double-sided VSD imaging to record the activity of all neurons in one ganglion (M10) within a short chain during a fictive bout of a reflexive behavior known as local bending, a withdrawal response to tactile stimulation in which the leech bends its body away from the stimulated location (*Kristan et al., 2005*). Fictive local bending can be induced readily even in isolated single ganglia or a short chain of ganglia by stimulating one of four pressure-sensitive sensory neurons (P cells).

Stimulating P cells causes a combination of excitation and inhibition in identified 'local bend inter-neurons' (LBIs) (*Kristan et al., 2005*; *Lockery and Kristan, 1990*). The LBIs synapse onto several motor neurons to produce an appropriate pattern of contraction and relaxation in the local area of the body wall that depends on which location (or which P cell) was stimulated (*Kristan et al., 2005*; *Lockery and Kristan, 1990*; *Baljon and Wagenaar, 2015*).

We induced fictive local bending by stimulating the left and right ventral P cells ($P_V^L$ and $P_V^R$) with trains of depolarizing pulses (20 Hz, 50% duty cycle, 1 s), which reliably evoked action potentials in those cells (*Figure 3a,b*). Intracellular recording from an Anterior Pagoda (AP) cell, which responds differentially to stimulation of ipsilateral vs. contralateral P cells (*Jin and Zhang, 2002*), was simulta-neously performed to confirm the different response to $P_V^L$ and $P_V^R$ stimulation and the correspon-dence of VSD and intracellular signals (*Figure 3b*). Stimuli were presented in order of LRRLLR…, for a total of 10 stimuli per P cell. From each of the resulting VSD traces, we extracted the average fluo-rescence change (ΔF/F) during the first 0.5 s of the stimulus as well as during a control phase (1–0.5 s before stimulus onset), both relative to a reference phase (0.5–0.1 s before stimulus onset; *Figure 3c*). Using a leave-one-out procedure, we calculated for each of the cells how reliably their activity could be used to 'predict' which of the P cells had been stimulated (*Figure 3d*). We then established a mapping between cells in the VSD images and identified neurons on the canonical maps to determine for all identified neurons to what degree their activity encoded stimulus identity (*Figure 3e*).

On average across eight experiments, 113 ± 11 (mean ± SD) cells on the ventral surface and 129 ± 6 on the dorsal surface could be mapped to identified neurons (*Figure 3f*). Among those, 28% of ventral cells [35 ± 11, mean ± SD] and 36% of dorsal cells (52 ± 18) encoded stimulus identity with prediction success higher than 75% during the first 0.5 s of the stimulus. This included one ven-tral LBI, all dorsal LBIs, and most motor neurons (MNs; *Figure 3g*). (All other ventral LBIs had predic-tion success in the range 65–75%. In contrast, the average prediction success in the control period was at chance level: 50.9%±1.3% (mean ± SEM) for both ventral and dorsal cells.) The other neurons with high prediction success were AP cells and Leydig cells, as well as cells provisionally identified as cells 56, 61, 251, and 152 on the ventral surface and cells 9, 10, 22, 28, 107, and 123 on the dorsal surface.

## Involvement of individual neurons in multiple behaviors

To further establish the utility of double-sided VSD imaging, we set out to determine to what extent neural circuit components are unique or shared between fictive three behaviors: local bending, swimming, and crawling. To do so, we evoked the corresponding fictive behaviors in isolated whole nerve cords using electrical stimulation (*Briggman and Kristan, 2006*). Specifically, local bending was activated by intracellular stimulation of a single $P_V^L$ or $P_V^R$ (*Kristan, 1982*); swimming was eli-cited by stimulating a DP nerve from either M11, M12, or M13; and crawling was elicited by stimulat-ing tail brain nerve roots. Motor patterns of fictive local bending and swimming were confirmed based on extracellular recordings of DP nerves or intracellular recording of AE cells (*Briggman and Kristan, 2006*; *Frady et al., 2016*; *Gu et al., 1991*). Crawling patterns were confirmed based on simultaneous intracellular recordings from two different motor neurons: the AE and CV cells (*Briggman and Kristan, 2006*). All three behaviors could be induced in each of six animals (*Vid-eos 1–4*).

We calculated the phase and magnitude of the coherence of each imaged neuron to the stimulus train (0.5 Hz) during fictive local bending; to the optical signal of motor neuron DI-1 during swim-ming; and to the intracellular trace of an AE cell during crawling. Results from all behaviors in one animal are shown in *Figure 4a–d* and *Videos 1–4*. Optical signals from representative cells located on both surfaces confirmed stereotyped activity patterns that were highly distinctive for each of the behaviors (*Figure 4e–h*).

We established identities of imaged neurons as before. On average over six preparations, we were able to assign 126 ± 11 cells on the ventral surface and 121 ± 10 on the dorsal surface. This allowed us to construct summary maps showing which neurons were consistently involved in which behaviors (*Figure 4j* and Materials and methods). Approximately, 10% of cells were involved in all three behaviors, 33% in two out of the three behaviors, 42% in a single behavior, and 9% in none of the three behaviors (*Figure 4j*). For the remaining 6% of cells, involvement in any of the behaviors could not be established due to lack of samples.

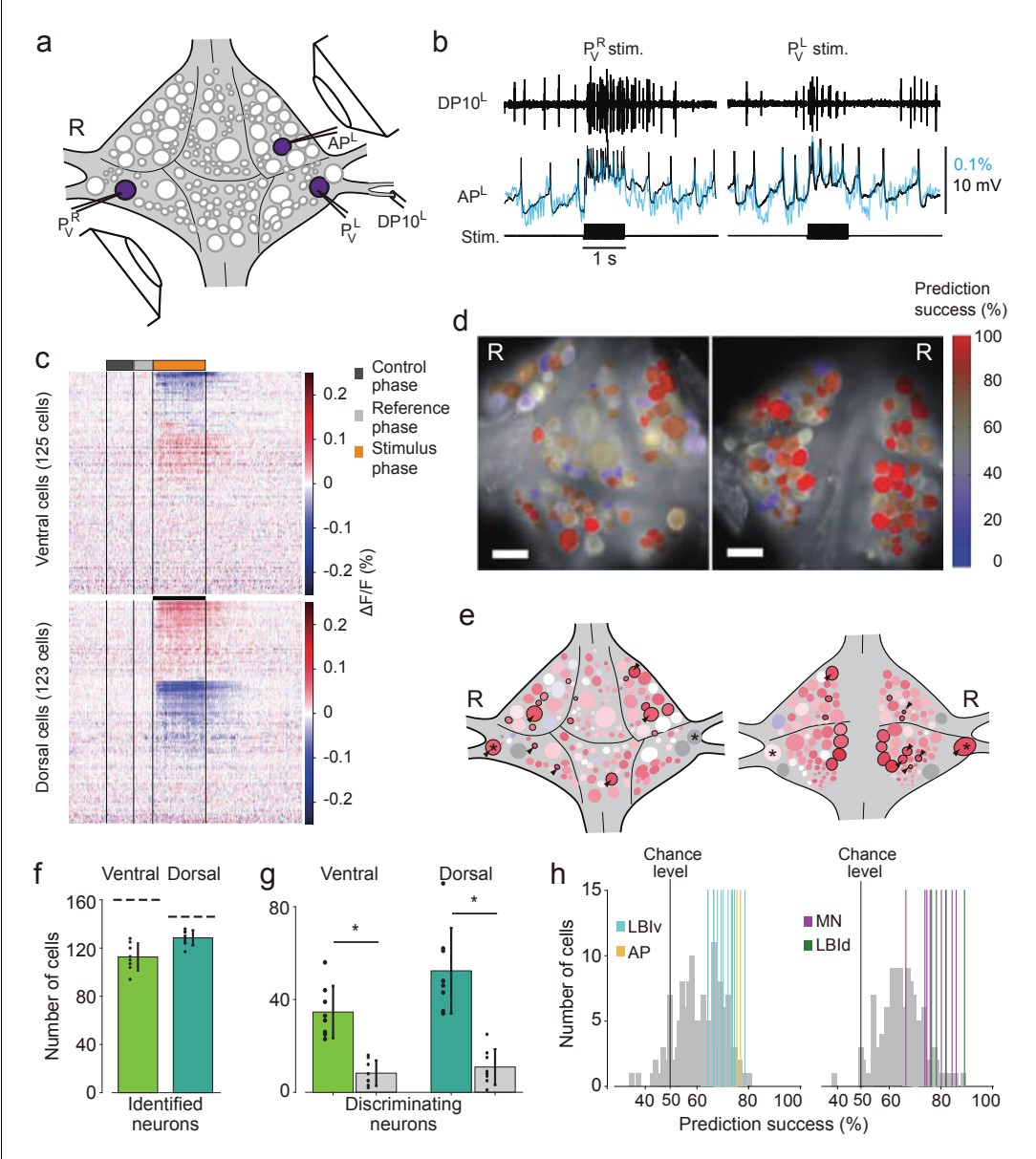

**Figure 3.** Differential activation during left and right local bend responses. (a) Schematic of the setup. Microelectrodes were inserted into left and right $P_V$ cells for stimulation and into the right AP cell for recording. A suction electrode around the right DP nerve confirmed the execution of a (fictive) local bend. (b) Simultaneously recorded motor activity from the DP nerve (*top*), membrane potential from the AP neuron (*middle, black*) and its corresponding VSD trace (*blue*) in response to stimuli to $P_V^R$ (*left*) and $P_V^L$ (*right*). Stimulus duration was 1 s (*bottom*). (c) Time series of averaged difference between $P_V^L$ (n = 10) and $P_V^R$ (n = 10) trials in the activity of all 248 recorded cells. Positive (red) indicates more depolarization (or less hyperpolarization) in response to $P_V^L$ stimulation. Scale bar: 1 s. (d) Stimulus discriminability score overlaid on images of the ventral (*left*) and dorsal (*right*) aspects of the ganglion. Scale bars: 100 µm. (e) Averaged discriminability results across eight animals. Color scale as in (d). Motor neurons (MNs) and LBIs are marked (*black circles*) as are other cells that strongly discriminate between stimuli (≥75% prediction success; *circles and arrow heads*). *: Leydig cell; see Discussion. (f) Number of cells that could be mapped to identified neurons; mean and SD of 8 preparations and individual results (*dots*). Dashed lines indicate total number of cells in the canonical maps. (g) Number of cells that strongly discriminate between stimuli (≥75% prediction success) compared to control (*grey bars*). (*: p<10⁻⁴; ventral: p=1.13×10⁻⁵, dorsal: p=4.72×10⁻⁵; Paired sample T-test) (h) Discriminability scores for all neurons on the ventral (*left*) and dorsal (*right*) surfaces. 50% prediction success represents a chance level. Colored lines mark the scores of LBIs, AP cells and MNs. Data available from the Dryad Digital Repository: https://doi.org/10.5061/dryad.m20kh/1 (title: Figure 3, *Tomina and Wagenaar, 2017*).

DOI: https://doi.org/10.7554/eLife.29839.009

The following figure supplement is available for figure 3:

**Figure supplement 1.** Mapping imaged cells to identified neurons using a graphical user interface.

*Figure 3 continued*

DOI: https://doi.org/10.7554/eLife.29839.010

Finally, we calculated a correlation matrix between the recorded activity of each of the cells, separately during each of the three fictive behaviors, and performed automated clustering based on these correlations (*Figure 5a*). For each of the cells in a recording, we then calculated what fraction of the cells in the same cluster was located on the ventral or the dorsal side of the ganglion. We found that during crawling and especially during local bending, most clusters were largely confined to only one side of the ganglion, whereas during swimming they more commonly spanned sides (*Figure 5b*), which indicates that swimming involves correlated activity among cells located on both surfaces whereas local bending largely does not. We quantified this by calculating an 'integration coefficient' (Materials and methods) which is equal to zero if all clusters are either wholly on the dorsal or wholly on the ventral side, and equal to one if all clusters are equally spread between the two sides (*Figure 5c*).

## Discussion

We constructed a double-sided microscope that can record fluorescence signals from two sides of a biological preparation. This technique should be broadly applicable to experimental questions that require simultaneous imaging from two widely spaced cell layers in *Drosophila* (*Kohsaka et al., 2017*), sea slugs (*Bruno et al., 2015*; *Hill et al., 2015*) and other organisms. The optical system can be assembled from conventional optic parts and devices. In our implementation, we used microscope parts from Olympus, but an equivalent system could be constructed using, for example, Thorlabs CERNA parts.

By combining our microscope with next-generation voltage-sensitive dyes (VF2.1(OMe).H (*Woodford et al., 2015*), we achieved simultaneous large-scale neuronal recording from two widely spaced cell layers at single-cell resolution, capturing not only action potentials but also small excitatory and inhibitory synaptic potentials. A primary feature of the system is its ability to acquire these signals at high speed, and without delay for image capture between the two focal planes. At present, this cannot be achieved by wide-brain volumetric $Ca^{2+}$ imaging as previously established for *C. elegans* (*Schrödel et al., 2013*; *Kato et al., 2015*). With our newly developed microscope, we simultaneously recorded, for the first time, the activity of the majority of neurons in a leech ganglion. While beyond the scope of this study, the fact that VSD recordings contain both spikes and postsynaptic potentials makes it possible to infer network connectivity among the different individual, identifiable cells. This offers a notable advantage over techniques that only give access to either spike events or intracellular $Ca^{2+}$ concentration.

The leech has 21 nearly identical segmental ganglia containing approximately 400 neurons that are arranged in a highly conserved geometry (*Kristan et al., 2005*). For 148 of these neurons, functional descriptions have been published. (A gateway to the relevant literature is available online, at http://www.danielwagenaar.net/ganglion.) The ganglionic neurons are distributed in a single layer on the surface of the ganglion, but this layer wraps around both the dorsal and ventral sides, so that at best half of the neurons can be simultaneously imaged with conventional microscopy. Our double-sided microscope, in contrast, has access to all of them, although surface curvature means that not all neurons can simultaneously be in sharp focus (*Figure 1b*). A single light source was sufficient for illuminating both top and bottom surfaces, because the leech nervous system is sufficiently

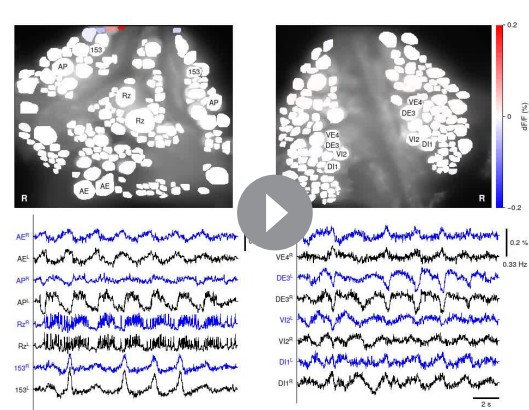

**Video 1.** Animated versions of *Figure 4a*, showing dynamics during $P_V^R$-induced local bending.
DOI: https://doi.org/10.7554/eLife.29839.011

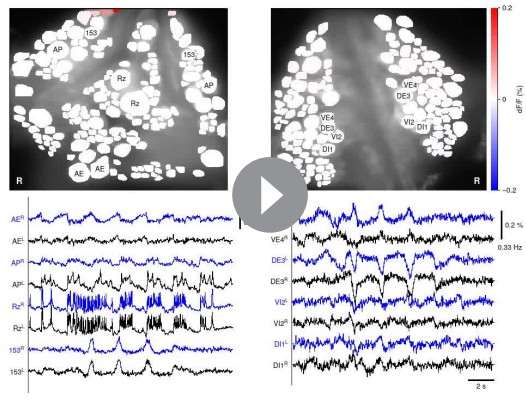

**Video 2.** Animated versions of *Figure 4b*, showing dynamics during $P_V^L$-induced local bending.
DOI: https://doi.org/10.7554/eLife.29839.012

translucent to permit even lighting onto both sides.

As typical in monopolar cells in invertebrate central nervous systems, the somata of leech ganglionic neurons have passive membrane properties and action potentials are electrotonically attenuated in the cell body from their origins in the neuropil. The somata of typical interneurons and motor neurons show small, attenuated action potentials. However, some other neurons, including sensory neurons and a few neurosecretory cells, exhibit larger, easily detectable spikes. In some neurons, unitary synaptic potentials are relatively easy to detect from the cell body because those potentials are not greatly attenuated. We could capture discrete unitary EPSPs (approximately 2–4 mV) as optical signals visually detectable even in single sweep of recordings (approximately 0.02–0.04% in $\Delta F/F$). In most neurons, synaptic interactions in the leech central nervous system during fictive behaviors usually result in compound synaptic potentials detectable even in the cell body. Hence, a low-noise imaging system using sensitive voltage sensors enables us to record behaviorally-relevant responses even in small neurons in the leech. In addition, our double-sided microscope is compatible with both intra- and extracellular electrode placement at least from one side, enabling detailed electrophysiological interrogation of selected specific neurons along with optical imaging from the whole population.

Intriguing features that we observed using our pan-neuronal imaging system are (1) widespread distribution of neurons that are differentially involved in left and right fictive ventral local bending (*Figure 3c,d*), and (2) involvement in multiple fictive behaviors of a large fraction of identifiable neurons (*Figure 4i,j*).

With respect to (1), we found that not only the local bend interneurons and the motor neurons previously reported (*Lockery and Kristan, 1990*) discriminated between the stimuli, but so did many other neurons that had not previously been implicated in fictive local bending. It has long been known that the neural mechanism of local bending involves population coding (*Lewis and Kristan, 1998a*; *Lewis and Kristan, 1998b*; *Lewis and Kristan, 1998c*; *Lewis, 1999*; *Thomson and Kristan, 2006*; *Kretzberg et al., 2016*), but its exact algorithm and computation remain unknown. Although the calculation of discriminability here was based on stimulus category ($P_V^L$ vs. $P_V^R$) instead of actual local bend patterns in the leech's body wall, the population dynamics of the highly

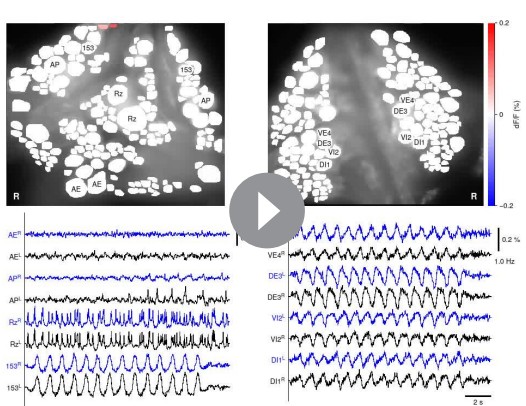

**Video 3.** Animated versions of *Figure 4c*, showing dynamics during fictive swimming
DOI: https://doi.org/10.7554/eLife.29839.013

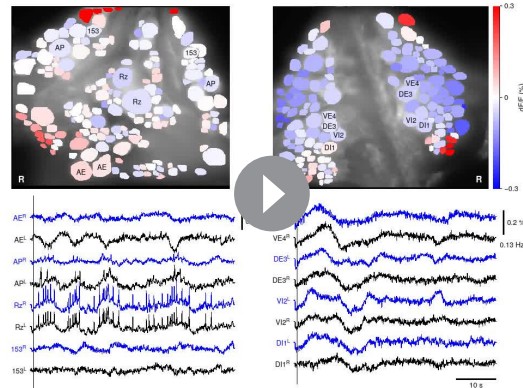

**Video 4.** Animated versions of *Figure 4d*, showing dynamics during fictive crawling.
DOI: https://doi.org/10.7554/eLife.29839.014

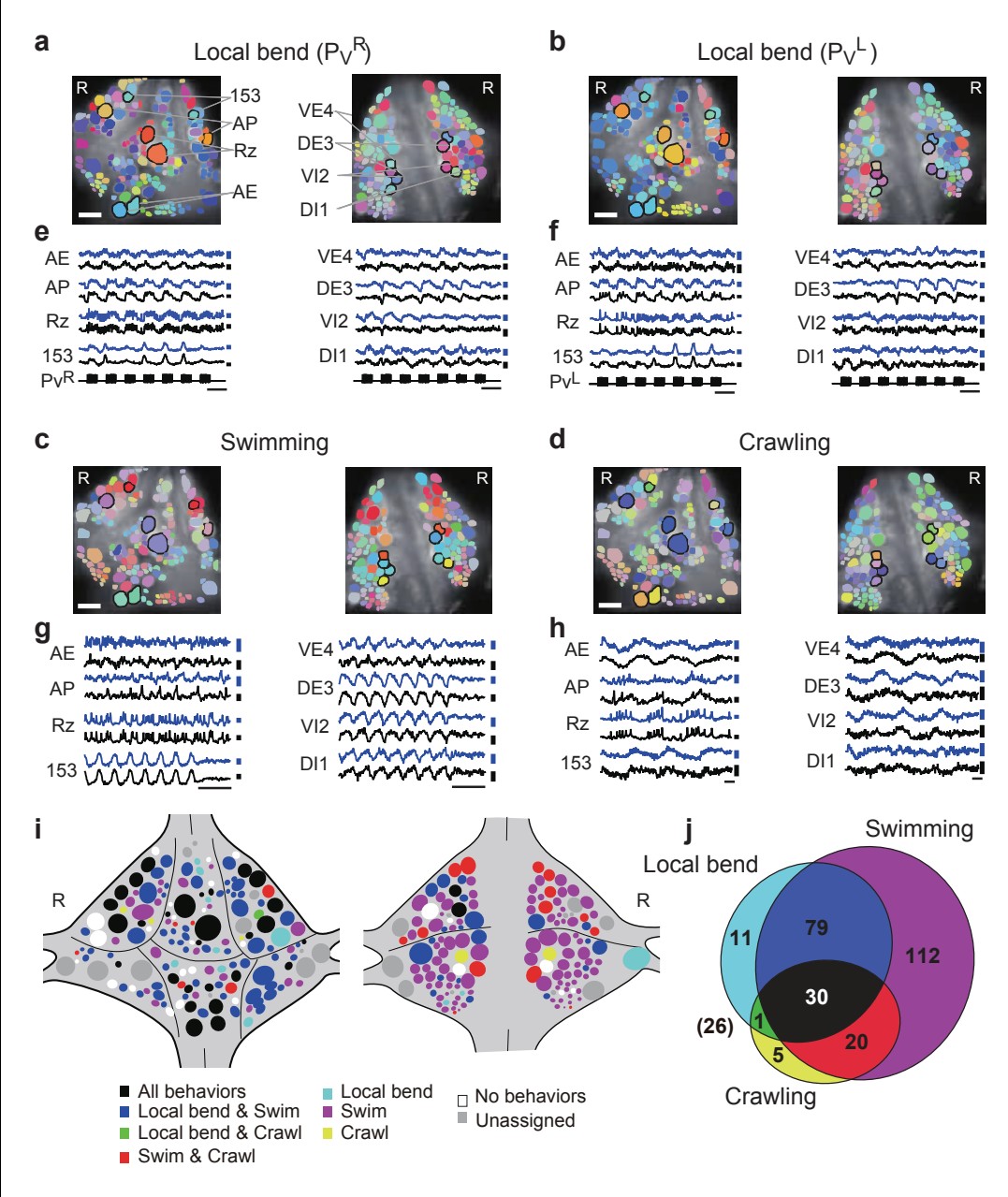

**Figure 4.** Neuronal activity during multiple behaviors. (a–d) Coherence of optically recorded signals of all cells on the ventral (*left*) and dorsal (*right*) surfaces of a ganglion with (a) $P_V^R$-induced local bending, (b) $P_V^L$-induced local bending, (c) fictive swimming, and (d) fictive crawling. Color map as in *Figure 2b*. (e–h) VSD signals of cells indicated in (a–d) during those behaviors. Scale bars: 2 s for time and 0.2% for ΔF/F. Blue and black traces represent the cells on the right and the left sides, respectively. (i) Summary maps of the involvement of identified neurons on the ventral (*left*) and dorsal (*right*) surface of the ganglion. Colors indicate which behavior each neuron was involved in. (j) Venn diagram showing the total number of identified neurons that oscillated with each individual behaviors or combinations of behaviors. The number (26) with parentheses outside of the diagram indicates the number of cells determined to be involved in none of the three behaviors. Colors as in (i). Data available from the Dryad Digital Repository: https://doi.org/10.5061/dryad.m20kh/2 (title: Figure 4, *Tomina and Wagenaar, 2017*).

DOI: https://doi.org/10.7554/eLife.29839.015

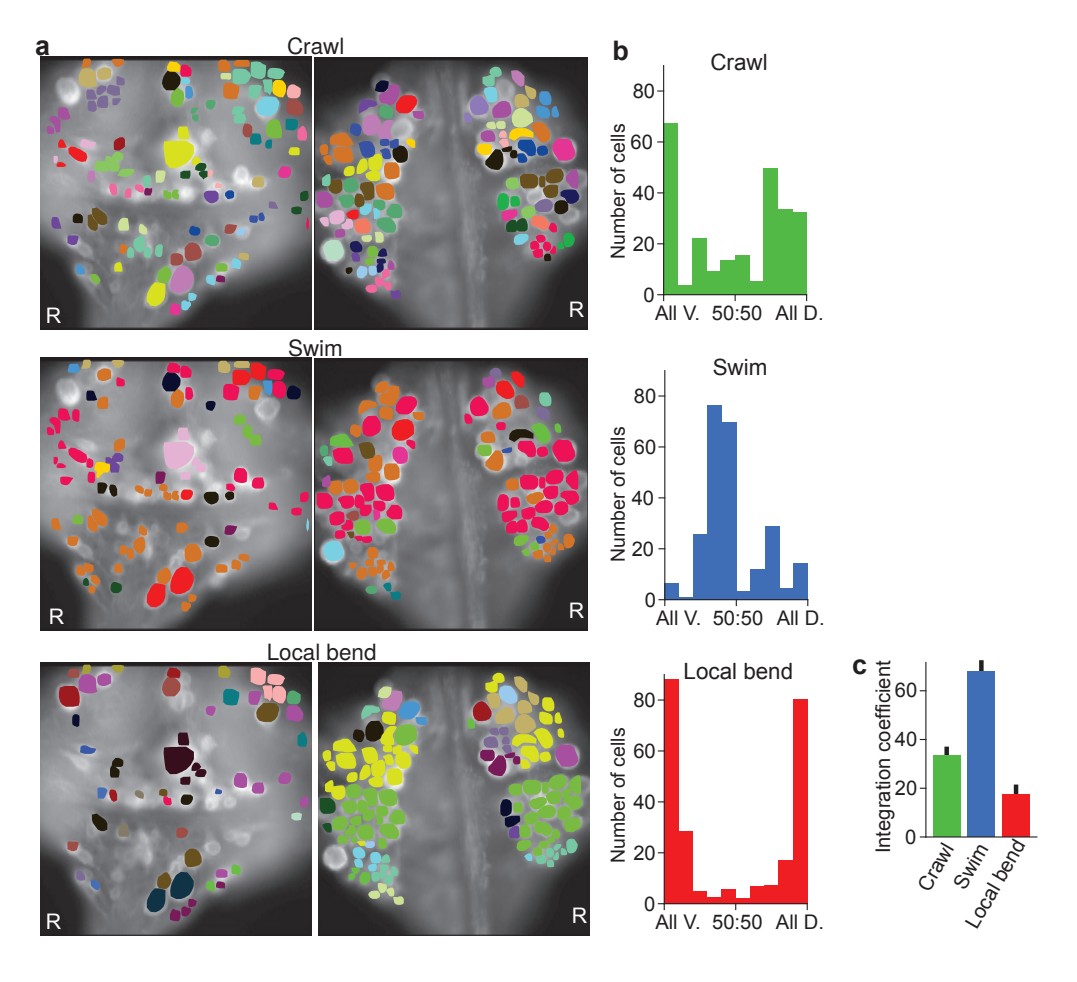

**Figure 5.** Clustering cells based on their activity in different behaviors. (a) Cluster assignments of all cells recorded in one animal based on the correlation matrix of their activity during fictive crawling (*left*), swimming (*center*), and local bending (*right*). (b) Degree to which cells within a cluster were fully contained on the ventral side ('All V.'), fully on the dorsal side ('All D.'), or equally distributed ('50:50'). To prevent overrepresentation of small clusters, each cell is an entry in the histogram, not each cluster. Clusters with fewer than three members were excluded. Data from N = 6 leeches. (c) Quantification of the degree to which members of clusters were distributed across surfaces in the three behaviors tested (mean ± SEM, N = 6). All differences were significant (ANOVA, $F(2,15) = 63.4$, $p<10^{-7}$, followed by Tukey). Data available from the Dryad Digital Repository: https://doi.org/10.5061/dryad.m20kh/3 (title: Figure 5, *Tomina and Wagenaar, 2017*).

DOI: https://doi.org/10.7554/eLife.29839.016

discriminative cells we identified putatively underlie the neuronal computation. The discriminability maps from our study can thus be utilized for future investigations of mechanosensory information processing: the maps will work as a guide to selectively record or manipulate by intracellular electrodes during local bend in order to assess the role of individual neurons on the behavior.

A few neurons can be seen from both sides of the ganglion, and some cells near the lateral edge of the ganglion may be cut off from the imaging. For instance, in *Figure 3e*, the right Leydig cell showed high discriminability and was seen in both dorsal and ventral aspects, but the left one was lost on the ventral side and had weak discriminability on the dorsal side. The main reason of such asymmetrical appearance on the discriminability maps could be explained as follows: We protected P cells from phototoxic effect by leaving sheath around the cell on the ventral side, thus the remaining sheath often covered other cells locating the lateral edge like Leydig cells. Because those cells are stained weakly or not at all especially on the ventral side, the asymmetrical results appeared.

With respect to (2), we observed that 43% of identifiable neurons on the ventral and dorsal surfaces were involved in at least two of the three behaviors tested (local bending, swimming, and

crawling). This result indicates that the neural circuits for those behaviors share many components while generating unique motor patterns for each behavior. The percentage of circuit components shared between swimming and crawling identified in this study differed from previous work (*Briggman and Kristan, 2006*); in particular, the number of cells we identified as involved in crawling (56) was lower than in the previous study (188). The reason is that double-sided desheathing, which is necessary for double-sided imaging, made long and regular episodes of fictive crawling relatively rare. Thus, crawl episodes in our experiments were somewhat shorter (typically only 3–4 cycles) than in the older study, resulting in a weaker coherence signal. In addition, we imaged for 50 s in our crawl trials, 10 s shorter than the previous study, so that we could capture 20 frames per second. This did result in less opportunity of inclusion of extended crawling oscillation cycles.

Double-sided imaging revealed a previously unappreciated difference between the swim rhythm and local bending: The cell assemblies that are simultaneously active in the former span both sides of the ganglion, whereas in local bending, they are mostly confined to either the dorsal or the ventral side. Whether this difference has any functional significance remains unknown.

In our study, the limited frame rate of the CCD camera (QuantEM 512SC; Photometrics) restricted imaging of brief action potentials. In the experiment for multiple behaviors, the frame rate for local bend and swimming was 50 Hz (for 15 s in duration), while the frame rate for crawling was 20 Hz (for 50 s), at the $512 \times 128$ spatial resolution adequate for making ROIs and for cell identification. In VSD imaging at 50 Hz frame rate, long lasting potentials (e.g. spikes in Retzius or Leydig cells) are easy to detect but brief action potentials (millisecond order) like an S-cell spike are hard to detect. When imaging at higher frame rate (e.g. 200 Hz as in the report of dye development [*Miller et al., 2012*]), the dye realizes sufficient reconstruction of such brief spikes. It will be necessary, then, to apply a higher frame rate than the current setting for connectivity analysis on detected spikes in future studies. Even in the present study, however, voltage imaging with VoltageFluor dye sufficiently reconstructed relatively large, slow action potentials in several cells (e.g. Leydig cells, Retzius cells and mechanosensory cells). This enabled us to confirm cell identity based on the characteristics of spontaneous firing activity and the shape of action potential and afterhyperpolarization.

The delay of the optical signal following voltage changes in VoltageFluor is negligible unlike with FRET-based VSDs because the speed of VoltageFluor is nanosecond to microsecond, while that of FRET dyes is millisecond to second (*Miller et al., 2012*). With VoltageFluor dyes (*Frady et al., 2016*; *Miller et al., 2012*; *Moshtagh-Khorasani et al., 2013*) including VF2.1(OMe).H (*Woodford et al., 2015*), we do not have to adjust phase delay which was commonly done to correct for the slow response feature of FRET-based dyes. For instance, in *Briggman and Kristan (2006)*, where they made single-sided coherence maps for fictive locomotory motor patterns, they used the slow dyes (FRET-based dyes) for imaging, using a frame rate of 10 Hz for swimming and 2 Hz for crawling (*Briggman and Kristan, 2006*). This likely explained why several cells like AE cells that show swim-related oscillation in our intracellular and VSD recordings (*Figure 2a*) were not detected in the their coherence analysis. By using a faster dye, we could obtain higher coherence in those neurons, especially during swimming.

In this study, we identified imaged cells with known neurons using a semi-automatic mapping algorithm based on cell size and location along with an expert's assessment based on the physiological properties of cells along with this geometrical information. To gain more insight into the neuronal networks responsible for behavior, it will be necessary to carry out more accurate neurocartography, which we will achieve by combining functional mapping using machine learning methods (*Frady et al., 2016*) with a connectomic approach using serial block face scanning electron microscopy (*Pipkin et al., 2016*). The combination of those techniques with double-sided VSD imaging will pave the way for future investigations on how the activity of all neurons in a central nervous system is recruited to process sensory information and to generate distinctive behaviors from overlapping neuronal circuits.

## Materials and methods

### Optical recording by double-sided microscope

We acquired fluorescence images simultaneously from two focal planes using a custom double-sided microscope consisting of the fluorescence train of an upright microscope (Olympus BX, Tokyo,

Japan) mounted on top of an inverted microscope (Olympus IX). The top microscope was used to image the upper focal plane while the bottom microscope imaged the lower focal plane. We used a 20x, 1.0 numerical aperture (NA) water-immersion objective, reduced to 0.7 NA by way of a custom aperture, for the upright and a 20x, 0.7 NA objective with cover-slip adjustment collar for the inverted microscope (both Olympus). The alignment of those two objectives was fine-adjusted manually so that cameras attached to the top and bottom microscopes saw the same field of view to within about 300 nm when the two focal planes were at the same depth.

The two objectives served as condenser for each other, so that blue excitation light delivered through the top objective for epifluorescence imaging also served as a transfluorescence light source for the bottom objective. Further, a red LED illuminator attached to the bottom microscope provided wide-field transillumination that enabled us to use the upright objective to visualize intracellular electrodes. Both objectives were mounted on standard turrets so that they could be rotated out of the way to make place for 5x objectives used to visualize extracellular suction electrodes.

For VSD imaging, we used excitation light (bandpass filtered to 470 ± 15 nm) from a high-power blue LED (LedEngin LZ1-10B200) controlled with optical stabilization (*Wagenaar, 2012*). In both the upright and inverted microscopes, we used a 490 nm dichroic mirror and 505 nm LP emission filter. Images were acquired with two cooled CCD cameras (QuantEM 512SC; Photometrics, Tucson, AZ) at a resolution of 512 × 128 pixels. The frame rate was set depending on which behavior was recorded: for local bending and swimming, images were acquired at 50 Hz; for crawling, images were acquired at 20 Hz. Imaging data were acquired using custom software VScope (*Wagenaar, 2017*). Optical and electrical recordings were synchronized by connecting frame timing signals from each camera to a data acquisition board that also recorded electrophysiology signals (see below).

VSD imaging is highly sensitive to even sub-micrometer motions. Because VSDs are located in cell membranes rather than the cytosol, a movement of less than 1% of a cell diameter can cause a signal change of well over 1% due to bright edge pixels moving out of a pre-defined region of interest (ROI). Since typical VSD signals are themselves far less than 1%, this can cause dramatic motion artifacts. To mitigate this problem, we replaced cooling fans inside each CCD camera with external blowers, since we determined that internal fans in cameras caused significant vibrations of the microscope objectives relative to the sample. After removing these fans, the remaining noise in image sequences was dominated by shot noise.

## Animal maintenance and sample preparation

Medicinal leeches (*Hirudo verbana*) were obtained from Niagara Leeches (Niagara Falls, NY) and maintained in artificial pond water at 15°C. For each experiment, an adult leech, regardless of its behavioral status at the time, was captured from an aquarium tank to make a preparation. The sample size of experiments of mechanosensory stimulus encoding (N = 8; *Figure 3*) and multiple behavioral generation (N = 6; *Figures 4* and *5*) was determined by referring to the previous study, by Briggman and Kristan, where single-sided VSD imaging was performed to obtain functional maps of the leech ganglion (*Briggman and Kristan, 2006*). In experiments where only local bending was the target behavior, we dissected out short chains of ganglia from segments 8 through 12. In experiments involving swimming or crawling, we isolated whole nerve cords, including the head brain, all 21 segmental ganglia, and the tail brain. In all cases, the blood sinus surrounding the nervous system was dissected away around segmental ganglion M10. We removed the sheath from the ventral and dorsal surface of this ganglion before applying voltage-sensitive dyes. To induce swimming, a dorsal posterior (DP) nerve root in one of ganglia M11 through M13 was stimulated through a suction electrode. Brief electrical pulses (3 ms) were delivered at 50 Hz in a 3-s-long train, with an amplitude of 7–8 V. To elicit crawling, several nerves from the tail brain were stimulated using the same stimulus parameters as for DP nerve stimulation. The preparation was put on a disk-shaped plate (diameter: 13 mm, thickness: 0.65 mm) with an open window (1.6 mm x 2.6 mm) on its center, made of PDMS (Sylgard 184, Dow Corning, Midland, MI). The target ganglion for imaging was placed over this window so that the PDMS substrate did not touch the bottom side of the ganglion and did not disturb focusing of the bottom objective. For imaging, we put this PDMS plate together with a preparation on the center of a glass-bottom dish (38 mm in diameter) whose periphery had PDMS substrate.

Whole nerve cords or short chains of leech ganglia can move slightly because muscle cells are embedded in the nerve cord. We therefore stabilized the ganglion to be imaged by tightly pinning

down blood sinus tissue to the PDMS substrate and by sandwiching adjacent connectives between small pieces of medical dressing (Tegaderm, 3M, Maplewood, MN), which was also pinned down, to minimize any motion artifacts. When imaging short chains of ganglia in local bend experiments, we disconnected inter-segmental interactions by pinching and crushing axon bundles inside of anterior/posterior connectives from M10 using forceps to reduce the ganglion's movement and variability of responsiveness of neurons in the ganglion. Throughout the dissection and during imaging, preparations were maintained in chambers filled with cold leech saline consisting of the following (in mM): 115 NaCl, 4 KCl, 1.8 CaCl2, 2 MgCl2, 10 glucose, and 10 HEPES, at pH 7.4. Only before crawling was induced, we temporarily replaced the cold saline (2–5°C) with room temperature (20–23°C) saline to obtain the most natural crawling rhythm. We bath loaded 800 nM VF2.1(OMe).H (*Woodford et al., 2015*) (provided by Evan Miller) in leech saline containing 1% pluronic acid (PowerloadTM Concentrate 100x, Thermo Fisher Scientific, Waltham, MA). To help with dye penetration into the cell membranes, we circulated the solution using a pair of peristaltic pumps (approximately 1.1 mL/min flow rate) with outflows directed at the dorsal and ventral surfaces of the ganglion, for 20 min total. The criteria for inclusion of preparation for data were (1) no obvious damage on neurons or connectives during desheathing or VSD staining was observed, and (2) a sufficient number of fictive behaviors was induced in a preparation for analysis.

We did not systematically examine when dye rundown occurs under certain level of exposure light brightness, but we could image fictive behavioral pattern with no major loss of imaging quality at least until the total light exposure reached at 250 s. For the left/right local bend experiment, single experiments usually took approximately 40 min of real time from the first trial, of which approximately 100 s was imaging time. For the multiple behaviors experiment, the time varied depending how smoothly all behaviors were successfully induced, but they typically took 30 min from the first trial, of which approximately 250 s was imaging time. According to *Miller et al. (2012)*, VoltageFluor dyes are less toxic than FRET-based dyes (*Miller et al., 2012*). Still, the combination of repetitive high-amplitude current injection (needed to reliably elicit multiple action potentials) and bright light exposure sometimes destroyed dye-stained P cells by causing injury bursting. That is why we protected P cells from phototoxicity by leaving sheath around it. Compared with P cells, dye-stained N cells were found to be relatively resistant to current injection and light exposure.

## Electrophysiology

We recorded intracellularly from up to three neurons simultaneously using 20–50 MΩ glass microelectrodes filled with 3 M potassium acetate and 60 mM potassium chloride, using Neuroprobe amplifiers (Model 1600; A-M systems, Sequim, WA). Intracellular recordings provided additional information regarding the behavioral state of the preparation as well as confirmation of the corresponding optical signals. We recorded extracellularly using suction electrodes and a four-channel differential amplifier (Model 1700; A-M Systems). All electrical signals were digitized at 10 kHz using a 16-bit analog-to-digital board (NI USB-6221; National Instruments, Austin, TX) and VScope software (*Wagenaar, 2017*).

## Basic data processing

We outlined the images of individual cell bodies manually as regions of interest using VScope (*Wagenaar, 2017*). Pixel values within each cellular outline were then averaged in each frame, yielding a raw fluorescence signal. Signals were processed to remove artifacts from micromotion (next section), and to correct for slow reduction of overall fluorescence intensity due to dye bleaching. The latter was achieved by subtracting locally fitted third-order polynomials using the SALPA algorithm (*Wagenaar and Potter, 2002*) with a time constant of 1 to 15 s. In addition, brightness averaged across the areas of the ganglion outside of ROIs was subtracted for each frame to reduce global noise due to fluorescent crosstalk among top and bottom images (*Lippert et al., 2007*). Finally, signals were normalized to their average value and expressed as a percent change in fluorescence ($\Delta F/F$).

## Motion correction

As mentioned above, motion artifacts were reduced by removing fans from CCD cameras and by pinning down ganglia tightly on the PDMS substrate. However, even very small motions can cause

highly detrimental artifacts in VSD recordings. To correct for small motions, we designated the middle frame of any recording as a reference frame, and generated a pair of artificial frames by shifting the reference frame one pixel to the left or to the right. Let $I_R$ and $I_L$ be vectors consisting of the intensity values of the pixels in the right- and left-shifted reference frames, and let $I'$ be the intensity vector of an arbitrary frame in the recording. As long as the motion is small (less than or approximately equal to one pixel),

$$\Delta x = 2(I' - I_L) \cdot (I_R - I_L) / \|I_R - I_L\|^2 - 1,$$

where $\cdot$ is the vector product and $\|I\|$ is the vector norm, is a good estimate for the motion in the x-direction between the frame under study and the reference frame. (The reason is that an image shifted by $\Delta x$ pixels can be approximated as

$$I' = [(1 - \Delta x)I_L + (1 + \Delta x)I_R]/2$$

as long as $|\Delta x| \lessapprox 1$. The first equation is derived from the second by minimizing with respect to $\Delta x$.)

The same method can of course be used for motion in the y-direction. More interestingly, the method can be used for other affine distortions as well. For instance, if we calculate artificial frames by rotating the reference frame by $\pm 0.1°$, the above procedure would yield estimates of image rotation (in units of $0.1°$).

Using this method, we estimated and corrected for small motions that may occur within the preparation or even due to vibrations in the microscope, thus preventing motion artifacts in the extracted VSD traces (*Figure 1—figure supplement 2*).

## Calculation of prediction success

In our experiments on the encoding of stimulus identity by individual neurons, we performed 10 trials stimulating the left $P_V$ cell and 10 stimulating the right $P_V$ cell, in order (LR)(RL)(LR)(RL)...Eight preparations with which we successfully performed 10 left Pv and 10 right Pv trials were collected in approximately 6 weeks. To calculate how well each cell 'predicted' the stimulus identity (i.e., 'left' or 'right'), we calculated the average $\Delta F/F$ during the first 0.5 s of each stimulus relative to the preceding reference phase, separately for each trial. Taking each trial in turn, we then took that trial and its 'partner' trial out, and calculated the average $\Delta F/F$ for the 9 'left' stimuli out of the remaining 18 trials and also for the 'right' stimuli. The 'partner' trial was the next trial for odd-numbered trials, and the preceding trial for even-numbered trials. If the $\Delta F/F$ in the trial under consideration was closer to the average $\Delta F/F$ of the 'left' trials in the training set than to the average of the 'right' trials, the neuron was considered correct in its 'prediction' of stimulus identity if the trial under consideration was in fact a 'left' trial, and conversely for 'right' trials. The percentage of trials in which a cell correctly predicted stimulus identity in this sense was used as a measure of prediction success. Any cell that correctly predicted stimulus identity in at least 75% of trials (50% being change performance) was considered to encode stimulus identity.

## Coherence analysis

We used multitaper spectral analysis (*Taylor et al., 2003*) to estimate the coherence between optical signals from individual cells with a common reference. That reference was the stimulus train for local bending, the optical signal of a DI-1 motor neuron for swimming, or the intracellular electrode signal of an AE motor neuron for crawling. For each recording, we calculated the 95% confidence interval for the magnitude of estimated coherence under the null hypothesis that a signal was not coherent with the reference (*Cacciatore et al., 1999*). A cell was considered to be involved in the behavior expressed during a given trial if its measured coherence exceeded this confidence interval.

## Canonical mapping

The overall layout of neurons within leech ganglia is highly conserved between ganglia within an animal as well as between animals, but the precise geometry does vary. In order to identify cells seen in the VSD image sequence (*Figure 3—figure supplement 1a*) with neurons in the canonical map, we developed a graphical user interface in GNU Octave (version 4.00) that allows us to proceed as follows. First, we mark all the visible cells as regions of interest on the image (*Figure 3—figure supplement 1b*). Then, we overlay the canonical map over this (*Figure 3—figure supplement 1c*). To the

trained eye, the identification of many of the larger cells is immediately obvious, so we register these identities (using a drag-and-drop mechanism in the GUI; (*Figure 3—figure supplement 1d*)). Cell identification in this step can be confirmed based on physiological properties observed in its optical signals. This partial mapping of ROIs to identified neurons allows the program to do a coarse alignment between the canonical map and the actual image using affine transformations local to each of the four packets of cells (*Figure 3—figure supplement 1e*). (The ganglion is divided by giant glial cells into six packets (*Kristan et al., 2005*), the boundaries of which are indicated on the canonical map.) This preliminary alignment enables us to identify several other neurons with high confidence, after which the computer can perform a local alignment step. Finally, the computer assigns putative identities to the remaining ROIs, leading to a nearly complete mapping between ROIs (orange dots in *Figure 3—figure supplement 1f*) and identified neurons (cross marks). This mapping technique can be applied to other preparations if a two-dimensional canonical map of any target aspect of the nervous system is available.

The following steps (1-3) of this procedure deserve further explanation.

## (1) Coarse alignment

For each aspect (ventral, dorsal) and each glial packet separately, the software has a list of (x,y)-coordinates of the canonical location of each neuron. Once the user has identified a subset of cells, a coarse alignment of the camera image with the canonical map is achieved by positing an affine transformation between camera coordinates $r = (X, Y)$ and canonical map coordinates $c = (x, y)$ so that— summed over all identified cells—the discrepancy between actual ROI centers ($r$) and mapped canonical centers $M(c)$ is minimized. The mapping $M$ is constructed as $M = T_2 \circ S_2 \circ R \circ S_1 \circ T_1$, where $T_1$, $T_2$ are translations, $R$ is a rotation, $S_1$ is an isotropic scaling matrix, and $S_2$ is an anisotropic scaling matrix. First, $T_1$ and $T_2$ are set so that the transformed positions of the canonical neurons and ROIs are centered around the origin, then $S_1$ is found to match the spreads of canonical and actual cells. Next, $R$ is found to match the rotation of the canonical and actual clusters, and finally $S_2$ is found to compensate for anisotropic stretching.

## (2) Fine alignment

A refinement of the coarse alignment is sought by positing a further local (nonlinear) translation $(\delta X, \delta Y)$ that is a function of location $(X, Y)$. Specifically,

$$\delta X(X, Y) = \sum \Delta X_k \, e^{-1/2\left((X-X_k)^2 + (Y-Y_k)^2\right)/\sigma_k^2}.$$

Here, $X_k, Y_k$ are the coordinates of the $k$-th identical ROI, $\Delta X_k$ is the mismatch between the actual x-coordinate of the $k$-th ROI and its prediction from $M(c_k)$, and $\sigma_k$ is a scale factor set to $\sqrt{3}$ times the distance between the $k$-th ROI and its nearest neighbor. This fine alignment ensures that all user-specified identities are respected geometrically, and that nearby cells are positioned in such a way to optimally respect local geometry.

## (3) Automatic assignment

After fine alignment, any canonical neurons that have not been selected by the user are mapped to the nearest ROI. When this would lead to conflicts (e.g., because two canonical neurons are both close to one ROI in the recording), these conflicts are resolved by considering matches between neurons and ROIs with similar sizes as most meritorious and by considering displacements of larger cells as less favorable than displacements of smaller cells.

## Determination of which cells are consistently involved in a behavior

For each neuron in each animal, we determined whether its coherence exceeded the 95% confidence interval of the null hypothesis that a given neuron was not involved in a given behavior. Six preparations with which we successfully induced the full set of behaviors (local bending, swimming and crawling) were collected in approximately 2 months. If a neuron exceeded that threshold for a given behavior in four out of six animals, it was considered to be involved in that behavior (*Briggman and Kristan, 2006*) (*Figure 4i*). Since swimming and crawling are both symmetric

behaviors, we included both members of a homologous pair if (and only if) at least one member exceeded the 97.5% C.I.

## Clustering and calculation of integration coefficients

We clustered cells based on the matrix of the correlation coefficients of their activity patterns, separately for each behavior (by constructing a dendrogram based on the correlation distance followed by tree cutting). We then assigned a dorsoventrality index (DVI) to each cell, which was equal to the fraction of dorsally located cells in that cell's cluster. This is what is shown in the histograms of *Figure 5b*. Cells in clusters with fewer than three members were ignored for this calculation; the results did not change qualitatively if this threshold was changed to two or five. Based on the DVI, we calculated the integration coefficient (CI) of *Figure 5c* as:

$$CI = \langle 1 - 2\,|DVI - 1/2|\rangle,$$

where $|\cdot|$ denotes absolute value and $\langle\,\cdot\,\rangle$ denotes the average across all cells (except those not in clusters of size three or more).

## Statistical analysis

All data processing and statistical analysis were performed in GNU Octave, version 4.0.0. Comparison of the number of stimulus-discriminating neurons between the stimulus phase and the control phase for *Figure 3g* was conducted using a paired sample t-test. Integration coefficients were compared among three fictive behaviors for *Figure 5c* using ANOVA followed by Tukey's test. Comparison of swimming parameters between single- and double-sided imaging for *Figure 2—figure supplement 1c–f* was carried out using ANOVA. Except where otherwise noted, the significance level was set at 0.05.

# Acknowledgements

We thank Evan Miller for sharing of the VF2.1(OMe).H dye; Annette Stowasser for her role in developing a prototype of the double-sided microscope and many helpful conversations; and Angela Bruno for useful discussions regarding data analysis. This work was supported by the Burroughs Welcome Fund through a Career Award at the Scientific Interface and by the National Institute of Neurological Disorders and Stroke through grant R01 NS094403 (both to DAW). YT was supported by JSPS Overseas Research Fellowships.

# Additional information

### Funding

| Funder | Grant reference number | Author |
|---|---|---|
| Japan Society for the Promotion of Science | JSPS Overseas Research Fellowships | Yusuke Tomina |
| National Institute of Neurological Disorders and Stroke | R01NS094403 | Daniel A Wagenaar |
| Burroughs Wellcome Fund | Career Award at the Scientific Interface | Daniel A Wagenaar |

The funders had no role in study design, data collection and interpretation, or the decision to submit the work for publication.

### Author contributions

Yusuke Tomina, Conceptualization, Data curation, Software, Formal analysis, Validation, Investigation, Visualization, Methodology, Writing—original draft, Project administration; Daniel A Wagenaar, Conceptualization, Resources, Data curation, Software, Formal analysis, Supervision, Funding acquisition, Validation, Visualization, Methodology, Project administration, Writing—review and editing

Author ORCIDs

Yusuke Tomina https://orcid.org/0000-0001-9406-1493
Daniel A Wagenaar https://orcid.org/0000-0002-6222-761X

Decision letter and Author response
Decision letter https://doi.org/10.7554/eLife.29839.020
Author response https://doi.org/10.7554/eLife.29839.021

## Additional files

### Supplementary files

• Transparent reporting form
DOI: https://doi.org/10.7554/eLife.29839.017

### Major datasets

The following dataset was generated:

| Author(s) | Year | Dataset title | Dataset URL | Database, license, and accessibility information |
|---|---|---|---|---|
| Tomina Y, Wagenaar D | 2017 | Data from: A double-sided microscope to realize whole-ganglion imaging of membrane potential in the medicinal leech | http://dx.doi.org/10.5061/dryad.m20kh | Available at Dryad Digital Repository under a CC0 Public Domain Dedication |

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
