## [Decision Letter]

Thank you for submitting your article "Whole-ganglion imaging of voltage in the medicinal leech using a double-sided microscope" for consideration by *eLife*. Your article has been favorably evaluated by Eve Marder (Senior Editor) and three reviewers, one of whom, Ronald L Calabrese (Reviewer #1), is a member of our Board of Reviewing Editors. The following individuals involved in review of your submission have agreed to reveal their identity: Farzan Nadim (Reviewer #2); Astrid A Prinz (Reviewer #3).

The reviewers have discussed the reviews with one another and the Reviewing Editor has drafted this decision to help you prepare a revised submission.

Summary:

This is an interesting manuscript describing a new a double-sided fluorescence microscope that can image two sides of a thin, relatively-translucent nervous system simultaneously. The microscope is compatible with recording electrodes advanced from the upper surface. The microscope was then applied to image neuronal activity in a single leech ganglion (using either short chains of ganglia or whole isolated nerve cords) using a state of the art VSD (VF2.1(OMe).H) from both surfaces so that potentially all cells are visible with single cell resolution. VSD recordings are supplemented with intracellular and extracellular recording to verify motor patterns and some cell identities; cell identities are largely inferred from soma size and positions from canonical maps. The data gathered in whole nerve cords with fictive local bends, fictive swimming, and fictive crawling were used to determine which of the identified neurons participate in each of the motor patterns and their overlap. They used the small chains of ganglia to determine if the recorded activity can discriminate sensory neurons that were stimulated to generate left vs. right fictive bends. The microscope has potential interest for all those working in thin, relatively-translucent nervous systems.

The work is carefully done with appropriate numbers of animals and within animal repeats, and the data analyzed appropriately. Necessary data and supplemental data and videos appears in easily accessible form.

Essential revisions:

1) There is some confusion about the data of Figure 4 and about how ganglia in the illustrations are oriented and right and left are assigned so that surfaces of the ganglia can be put together in a coherent way. Please see reviewer #1's detailed comments.

2) There is concern that the mapping technique is not adequately described and that it will not be applicable to other preparations in which cell bodies are not so stereotypically arranged. Does this limit the potential use of the microscope?

3) The advantages of using a fast VSD are emphasized but the advantage is not apparent in the data presented or its interpretation. Moreover, frame rate, dye rundown, and dye toxicity may limit experiments more than is implied. Please see reviewer #1's detailed comments.

*Reviewer #1:*

1) Figure 4 is critical and I don't really understand it because it is not fully explained in the text or legend. As far as I can understand from Materials and methods etc. you are looking at left-right difference to tell discrimination. Ok, then all cells lit up should be bilateral pairs as the left cell should be different from the right to discriminate PL from PR. Is this correct? What does it mean that only one Leydig cell discriminates? I assume that the Leydig R is visible on both ganglionic faces; but see note below.

(I am not sure that I understand how the ganglia are oriented and you don't say in the legend. Is the ventral surface shown in true ventral view? I.e., will I rotate the ventral view on its long axis one half rotation to map it onto the dorsal view? For example is that the Leydig cell R on the left side of the ventral view. This is very important and needs to be specified for *all* figures. I thought that it was standard leech practice to call L and R with respect to the animal oriented dorsal side up – this is how humans do it for other humans; on the canonical map of a leech ganglion in the ventral view R is on the left of the illustration and L is on the right or have I been doing it wrong for 35 years? Under this rule then the map of Figure 4 is labeled incorrectly and sows confusion about how we are to map the surfaces on to one another. Why don't you just label your ganglia with L and R (absolute, i.e. with respect to a dorsal side up animal) and have done with the confusion?)

2) You make a big point of using a fast VSD but your phase results ("Using the VF2.1(OMe).H dye, we were able to confirm the oscillatory behavior of neurons previously studied using an earlier-generation dye. In addition, we were able to detect weaker oscillations in many other neurons on both sides of the ganglion.") were not different from previous findings. Then in Discussion you say "The percentage of circuit components shared between swimming and crawling identified in this study differed from previous work; in particular, the number of cells we identified as involved in crawling was lower than in the previous study (188). The reason is probably that crawl episodes in our experiments were somewhat shorter (typically only 3-4 cycles) than in the older study, resulting in a weaker coherence signal." You also make claims about being able to decipher connectivity because you use a fast sensitive dye. Then in Materials and methods you give us the video rate as "… for local bending and swimming, images were acquired at 50 Hz; for crawling, images were acquired at 20 Hz." These inconsistencies raise a number of questions. How have you advanced the field by using a fast VSD? What new have we learned about phase relations or circuit operation for swimming, e.g., because you used a fast VSD? Is connectivity really possible with 50Hz video? Certainly not with 20 Hz video. Did the ability to detect spikes really add to your analyses? Why were crawl episodes shorter in your study, and why did not greater dye sensitivity allow you to make up for the smaller number of cycles? How long can you image in your set up without preparation or dye rundown? How long is a typical imaging run? What is known dye toxicity?

---

## [Author Response]

Essential revisions:1) There is some confusion about the data of Figure 4 and about how ganglia in the illustrations are oriented and right and left are assigned so that surfaces of the ganglia can be put together in a coherent way. Please see reviewer #1's detailed comments.

Indeed, the orientation and the left/right assignment of the leech ganglia in figures should be presented in the correct way as reviewer #1 indicated. We dealt with this issue by fixing the orientation of dorsal images of the ganglion in Figure 1–Figure 5, Figure 2—figure supplement 1 and Figure 3—figure supplement 1 and Video 1–Video 4, adding “R” marks in the figures, and also correcting expression of left/right in the revised manuscript as reviewer #1 pointed out in the latter part of point #1.

2) There is concern that the mapping technique is not adequately described and that it will not be applicable to other preparations in which cell bodies are not so stereotypically arranged. Does this limit the potential use of the microscope?

We added further explanations for the canonical mapping technique, especially about course alignment, fine alignment and automatic assignment, in the Materials and methods section (subsection “Canonical mapping”).

The mapping technique will be applicable to other preparations if there is a two-dimensional canonical map of any target aspect of the nervous system. Regardless of how much variety exists in a certain neuronal preparation (a brain or a ganglion), it will be necessary to prepare and register a canonical map to overlay over an obtained image to match ROIs and canonical cells if a researcher wants to apply the mapping method. If experimenter can visually identify a few cells based on its cell size, position, or shapes and also confirm it by physiological features, the following procedures (course alignment, fine alignment and automatic assignment) would more effectively work for increasing success probability of the canonical mapping. There are more potential applicable ways of the double-sided microscope other than comprehensive canonical cell identification. Even in a preparation in which cell bodies are not well stereotypically arranged, the microscope will be able to work to characterize multiple cells whose function is unknown, or to analyze neural population dynamics, based on subthreshold postsynaptic activities.

3) The advantages of using a fast VSD are emphasized but the advantage is not apparent in the data presented or its interpretation. Moreover, frame rate, dye rundown, and dye toxicity may limit experiments more than is implied. Please see reviewer #1's detailed comments.

We dealt with this issue by adding explanations to those issues in the Discussion and the Materials and methods, for the issue of VoltageFluor’s advantages (Discussion, eleventh paragraph) and for frame rate (Discussion, tenth paragraph), dye rundown and dye toxicity ((subsection “Animal maintenance and sample preparation”, last paragraph),) in the revised manuscript as the reviewer #1 pointed out in point #2.

Reviewer #1:1) Figure 4 is critical and I don't really understand it because it is not fully explained in the text or legend. As far as I can understand from Materials and methods etc. you are looking at left-right difference to tell discrimination. Ok, then all cells lit up should be bilateral pairs as the left cell should be different from the right to discriminate PL from PR. Is this correct? What does it mean that only one Leydig cell discriminates? I assume that the Leydig R is visible on both ganglionic faces; but see note below.(I am not sure that I understand how the ganglia are oriented and you don't say in the legend. Is the ventral surface shown in true ventral view? I.e., will I rotate the ventral view on its long axis one half rotation to map it onto the dorsal view? For example is that the Leydig cell R on the left side of the ventral view. This is very important and needs to be specified for all figures. I thought that it was standard leech practice to call L and R with respect to the animal oriented dorsal side up – this is how humans do it for other humans; on the canonical map of a leech ganglion in the ventral view R is on the left of the illustration and L is on the right or have I been doing it wrong for 35 years? Under this rule then the map of Figure 4 is labeled incorrectly and sows confusion about how we are to map the surfaces on to one another. Why don't you just label your ganglia with L and R (absolute, i.e. with respect to a dorsal side up animal) and have done with the confusion?)

The asymmetric results in the discriminability maps for Pv^L/R^ stimulation in Figure 3, especially for lateral large cells like the pair of Leydig cells, is due to cell missing in an imaging experiment. Main reason of such cell missing is that we left sheath on the lateral side around Pv cells not to stain them. Repetitive intracellular current injection and light exposure sometimes killed P cells during imaging trial. To avoid this, we protected the target P cells from dye by leaving ventral sheath around the lateral nerves. This made some cells near to Pv cells occasionally not enough stained or invisible especially from ventral side. Because those cells are often unstained or missing, the asymmetric results easily appears. In fact, in Figure 3, the right Leydig cell appeared to show high discriminability and the left one is missing in ventral side, while the right same cell showed high discriminability and the left one had weak discriminability on dorsal side. We mentioned it in the revised manuscript (Discussion, seventh paragraph).

As reviewer#1 pointed out, we agree to label ganglia with R in the absolute manner for all relevant figures and added an explanation of the ganglion’s orientation in the relevant figures. In the old manuscript, the orientation of the dorsal images (Figure 1–Figure 5, Figure 2—figure supplement 1 and Figure 3—figure supplement 1) from the bottom camera is not appropriate because those images are reflections. We fixed those figures by flipping around as mirror image. In figure legend of Figure 1, we added an explanation of this left/right issue. Also, in the revised manuscript, we corrected the expression of left/right if that was inappropriate.

2) You make a big point of using a fast VSD but your phase results ("Using the VF2.1(OMe).H dye, we were able to confirm the oscillatory behavior of neurons previously studied using an earlier-generation dye. In addition, we were able to detect weaker oscillations in many other neurons on both sides of the ganglion.") were not different from previous findings. Then in Discussion you say "The percentage of circuit components shared between swimming and crawling identified in this study differed from previous work; in particular, the number of cells we identified as involved in crawling was lower than in the previous study (188). The reason is probably that crawl episodes in our experiments were somewhat shorter (typically only 3-4 cycles) than in the older study, resulting in a weaker coherence signal." You also make claims about being able to decipher connectivity because you use a fast sensitive dye. Then in Materials and methods you give us the video rate as "… for local bending and swimming, images were acquired at 50 Hz; for crawling, images were acquired at 20 Hz." These inconsistencies raise a number of questions.How have you advanced the field by using a fast VSD? What new have we learned about phase relations or circuit operation for swimming, e.g., because you used a fast VSD?

Although there is such limitation of spike detection for analysis due to the lower frame rate, VSD imaging with this fast dye reconstructed relatively large, slow action potential’s trace sufficiently in several cells (e.g. Leydig cells, Retzius cells and mechanosensory cells). It helped us to confirm cell identification based on characteristics of spontaneous firing activity, spike response, and afterhyperpolarization. With VoltageFluor dyes including VF2.1(OMe).H, the delay of optical signal to voltage change is negligible unlike the case of FRET-based VSD with slow speed. The speed scale for VoltageFluor is nanosecond to microsecond, while that of FRET dye is millisecond to second (Miller et al., 2012). Unlike FRET-based dyes, we do not have to adjust phase delay, which is caused by slow response feature of dye, using the fast dye.

In Briggman & Kristan (2006) where they made coherence maps single-sidedly for fictive swimming and fictive crawling, they used slow dyes (FRET-based dyes) for imaging. The frame rate was 10 Hz for swimming and 2 Hz for crawling. (In 10926 of the previous report, they stated “For swimming, images were acquired at 20 Hz, yielding ratioed signals at 10 Hz. Trials at this frame rate were 10 s in duration. For crawling, 50 ms frames were acquired twice per second, yielding 2 Hz ratioed signals.”). In their results on fictive swimming, several cells like AE cells which show swim-related oscillation in intracellular recording (clearly shown in Figure 2) were not detected in the coherence map. By using the fast dye and imaging at 50 Hz, we could obtain higher coherence in those neurons especially for swimming. We mentioned it in the revised manuscript (Discussion, tenth and eleventh paragraphs).

Is connectivity really possible with 50Hz video? Certainly not with 20 Hz video. Did the ability to detect spikes really add to your analyses?

We did not apply spike detection and analysis of spikes. In our study, frame rate of imaging is the main limitation for detecting action potentials. If we perform imaging at enough frame rate with the fast VSD (e.g. sampling rate: 200 Hz in Miller et al., 2012), the dye realizes sufficient reconstruction of spike shape even in mammalian cell showing brief action potentials. In our experiment for multi-behaviors analysis, the frame rate for local bend and swimming was 50 Hz for 15 sec in duration, while the frame rate for crawling was 20 Hz for 50 sec. In 512 x 128 spatial resolution, those frame rate settings were the maximum limitations for those recording duration in our CCD cameras (QuantEM 512SC; Photometrics). In VSD imaging at 50 Hz frame rate, a brief action potential (a millisecond order) like S cell spike is hard to detect, while a long lasting action potential like Leydig cell spike (30 millisecond) is easy to detect. Therefore, if we apply spike detection to the VSD imaging data using the current apparatus, we cannot avoid a bias based on the duration of action potentials. It is necessary to apply higher frame rate than the current setting for connectivity analysis based on detected spikes from VSD data in future, and it is difficult to perform such analysis with VSD data obtained by the current setting of our CCD cameras.

In the previous study using VoltageFluor in single side of the ganglion (Moshtagh-Khorasani et al., 2013), they imaged spontaneous activities (not behavior) and analyzed spike trains detected from the VSD traces to describe spike cross-correlation pattern in 14 neurons for ventral and 27 neurons for dorsal. They performed VSD imaging at a sampling rate from 94 to 110 frame/s and at a spatial resolution of 128 x 128 pixels with Electron Multiplier CCD camera (C9100-13, Hamamatsu). They used 2~5 times faster frame rate than ours so that they could detect more spikes than ours. However, they lost spatial resolution (512 x 128 in our imaging) so that it was not good for drawing ROIs especially for small neurons. We could have performed a similar experiment with the same setting (faster sampling rate, lower spatial resolution) to see spike cross-correlation of spontaneous activities or some behaviors. In such situation, the quality of drawling ROIs and identification of cells would, naturally, be as not good as that in our current study. Therefore, we may have obtained several large identified cells to be included in spike cross-correlation across ventral and dorsal sides.

Why were crawl episodes shorter in your study, and why did not greater dye sensitivity allow you to make up for the smaller number of cycles?

Coherence degree largely depends on the cycle of fictive behavior under recording time. Even in 15 sec recording, swim cycle is usually sufficient to obtain significant coherence degree because fictive swimming typically continues for several seconds and its frequency is relatively fast (approximately 1-1.5 Hz). Compared with fictive swimming, fictive crawling is slower event (approximately 0.1 Hz) and is relatively difficult to observe as long-lasting regular rhythmic manner at least in the double-desheathed preparation. Double sided desheathing might cause unhealthy effect on the ganglion preparation especially for maintenance of fictive crawling bout. Recording time for crawling was 50 sec (at 20 Hz), which is 10 sec shorter than the previous study, also causing less opportunity of inclusion of crawling oscillation. VoltageFluor dyes are faster than FRET based dyes in response speed, but VoltageFluor’s sensitivity is not greater than that of FRET based dyes. Sensitivity of VF2.1(OMe). H. in the leech ganglion we used was 2.7%, but that of FRET-based dye was 4.3% according to Cacciatore et al. (1999). We mentioned it in the revised manuscript (Discussion, eighth paragraph).

How long can you image in your set up without preparation or dye rundown? How long is a typical imaging run?

We did not systematically examine when dye rundown caused problematic effect during imaging at certain brightness of exposure light, but we could image fictive behavioral pattern at least even 250 sec after excitation light exposure. The toughness of preparation to imaging experiment largely depends on its health condition of preparation. In order to keep its health condition well, an experimenter needs to finish dissection process quickly and frequently replace saline with cooler one during the process.

For the left/right local bend experiment, single experiment usually took about 40 min including inter-trial intervals from the first trial. The running time of imaging was about 100 sec in total (5 sec for each trial, 20 trials for each experiment). For the multi-behaviors experiment, the time varied depending how smoothly all behaviors were successfully induced, but it typically took 30 min including inter-trial intervals from the first trial. The running time of imaging was up to about 250 sec. We mentioned it in the revised manuscript (subsection “Animal maintenance and sample preparation”, last paragraph).

What is known dye toxicity?

We found only single relevant explanation regarding toxicity of this type of dyes in Miller et al. (2012) where the first version of VoltageFluor, VF2.1.Cl was reported. In P.2117, there is a statement that “Tests of the toxicity and bleaching of the PeT-based VSD similar to those performed on the FRET-based dyes show that the PeT-based VSD has a slower rate of bleaching and is less toxic than the FRET-based dyes.”, but they did not show data of its toxicity. As far as we know based on our experience, the combination of repetitive, strong current injection and strong excitation light exposure sometimes destroys especially stained P cells by causing injury bursting. Compared with P cells, N cells are relatively tough to current injection and light exposure under stained condition. We think that its photo-toxicity depends on cells, although we are not sure what properties of individual cells are related to tendency of death to the toxicity. We mentioned it in the revised manuscript (subsection “Animal maintenance and sample preparation”, last paragraph).